**Atmospheric chemical processing dictates aerosol aluminum solubility: insights from field measurement at two locations in Northern China**

Tianyu Zhang,[1,5] Yizhu Chen,[1,5] Huanhuan Zhang,[2] Lei Liu,[3] Chengpeng Huang,[4] Zhengyang Fang,[1,a] Yifan Zhang,[1,5] Fu Wang,[4] Lan Luo,[4] Guohua Zhang,[1] Xinming Wang,[1] Mingjin Tang[1,6,*]

[1] State Key Laboratory of Advanced Environmental Technology and Guangdong Key Laboratory of Environmental Protection and Resources Utilization, Guangzhou Institute of Geochemistry, Chinese Academy of Sciences, Guangzhou, China

[2] Guangzhou Marine Geological Survey, China Geological Survey, Guangzhou, China

[3] Hangzhou International Innovation Institute, Beihang University, Hangzhou, China

[4] Longhua Center for Disease Control and Prevention of Shenzhen, Shenzhen, China

[5] College of Earth and Planetary Sciences, University of Chinese Academy of Sciences, Beijing, China

[6] Institute of Surface-Earth System Science, School of Earth System Science, Tianjin University, Tianjin, China

[a] Current address: Institute of Low Temperature Science, Hokkaido University, Sapporo, 060-0819, Japan

Correspondence: Mingjin Tang (mingjintang@126.com)

**Abstract**

Deposition of mineral dust aerosol into open oceans impacts marine biogeochemistry, and the deposition flux can be constrained using dissolved aluminum (Al) in surface seawater as a tracer. However, aerosol Al solubility, a critical parameter used in this method, remains highly uncertain. We investigated seasonal variations of aerosol Al solubility for supermicron and submicron particles at two locations (Xi'an and Qingdao) in Northern China. Aerosol Al solubility was very low at Xi'an, showed no apparent variation with seasons or relative humidity, and was not correlated with sulfate or nitrate; in contrast, Al solubility was much higher at Qingdao, exhibited distinct seasonal variability, and increased with relative humidity and the abundance of sulfate and nitrate. All these features observed for Al solubility at the two locations can be explained by the effects of atmospheric chemical processing. Mineral dust transported to Xi'an (an inland city in Northwest China) was still not obviously aged and thus chemical processing had little effect on aerosol Al solubility; after arriving at Qingdao (a coastal city in the Northwest Pacific), mineral dust was substantially aged by chemical processing, leading to significant enhancement in aerosol Al solubility. Our work further reveals that aerosol liquid water and acidity play vital roles in the dissolution of aerosol Al by atmospheric chemical processing. We suggest that chemical aging can lead to spatiotemporal variation of aerosol Al solubility, and this should be considered when using dissolved Al in surface seawater to constrain oceanic dust deposition.

## 1. Introduction

As an important type of tropospheric aerosols, mineral dust aerosol greatly impacts atmosphere chemistry, climate, and ecological systems (Jickells et al., 2005; Tang et al., 2016; Kok et al., 2023). After long-range transport, deposition of mineral dust into the oceans is a major external source of several nutrient and toxic elements for surface seawater (Moore et al., 2013; Westberry et al., 2023), impacting primary production and biogeochemical cycles in the oceans and having further feedback on the climate system (Mahowald, 2011; Jiang et al., 2024). The deposition flux of mineral dust aerosol into the oceans should be accurately estimated before we can assess its impacts on marine biogeochemistry in a reliable manner (Schulz et al., 2012; Anderson et al., 2016). Previous studies used several different methods to estimate dust deposition fluxes and found large discrepancies (Huneeus et al., 2011; Anderson et al., 2016).

Deposition of mineral dust aerosol is the dominant source of dissolved aluminum (Al) in the surface water of open oceans, and dissolved Al is generally considered to be chemically and biologically inactive in seawater. As a result, dissolved Al concentrations in surface seawater could be used to calculate dust deposition flux into the oceans (Measures and Brown, 1996; Measures and Vink, 2000), and the fractional solubility of aerosol Al (the fraction of aerosol Al that can be dissolved) is one of the key parameters used in this method. Previous studies which used this method to estimate dust depositions fluxes (Han et al., 2008; Measures et al., 2010; Grand et al., 2015; Benaltabet et al., 2022) usually assumed uniform Al solubility values in the range of 1.5-5%. However, field measurements found that aerosol Al solubility could vary by more than an order of magnitude (Baker et al., 2006; Buck et al., 2013), and thereby using a uniform aerosol Al solubility value could lead to large uncertainties in

estimated dust deposition fluxes (Han et al., 2008; Xu and Weber, 2021). In order to better
constrain the oceanic dust deposition using dissolved Al in seawater as a tracer, we need to
develop parameterizations for aerosol Al solubility, and this requires spatiotemporal variability
of aerosol Al solubility to be understood and processes and mechanisms which drive such
variations to be elucidated.
The initial Al solubility is generally low (typically <1.5%) for soil or mineral dust samples
(Mulder et al., 1989; Duvall et al., 2008; Shi et al., 2011; Aghnatios et al., 2014; Li et al., 2022),
and field studies found that aerosol Al solubility in the troposphere could be much higher and
showed wide variability. For example, Al solubility ranged from 0.2-15.9% for total suspended
particles (TSP) over the Pacific (Buck et al., 2013), and were in the range of 3-78% over the
Atlantic (Buck et al., 2010; Chance et al., 2015). Some studies (Measures et al., 2010; Sakata
et al., 2023) found good correlations between dissolved aerosol Al (or Al solubility) and acid
species in aerosol particles, and thus suggested that chemical processes in the atmosphere could
substantially enhance aerosol Al solubility; furthermore, Li et al. (2017) found that Al solubility
was remarkably increased during cloud events when cloud processing enhanced the formation
of secondary inorganic ions (mainly sulfate and nitrate) and thus increased the acidity of cloud
droplets. However, Yang et al. (2023) found no correlations between Al solubility and the
concentrations of aerosol acidic species, and concluded that the effect of acid processing on Al
solubility was negligible. Aerosol Al solubility over the Atlantic appeared to be higher for air
masses from Europe than those from the Saharan region (Baker et al., 2006; López-García et
al., 2017), and some studies hypothesized that this could be potentially explained by the
influence of anthropogenic aerosol Al if it had higher solubility than mineral dust (Paris et al.,
2010; López-García et al., 2017).

It can be concluded that although aerosol Al solubility in the atmosphere was explored by

several previous studies, our understanding is still very limited. For example, it remains unclear
why aerosol Al solubility shows large spatial and temporal variation. Some work suggested
that atmospheric chemical aging could enhance aerosol Al solubility, but the mechanisms and
key environmental factors have not been elucidated. Furthermore, the effects of particle size
on aerosol Al solubility have not been well understood.

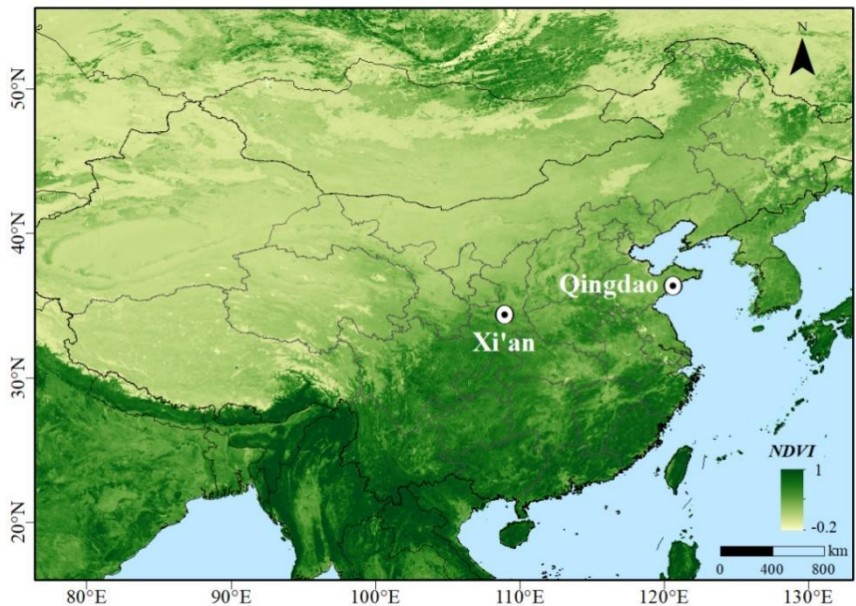


**Figure 1.** A map of East Asia and surrounding areas. The two locations (Xi'an and Qingdao)
where we collected aerosol particles are highlighted. NDVI: normalized difference vegetation
index provided by MODIS (Moderate Resolution Imaging Spectroradiometer).

In this work, we collected supermicron (>1 μm) and submicron (<1 μm) aerosol particles

at Xi'an and Qingdao, both located in Northern China, and investigated seasonal variations of
aerosol Al solubility at these two locations. Taklimakan and Gobi Deserts in Northwestern
China are two important source regions of Asian dust (Prospero et al., 2002). As shown in
Figure 1, Xi'an is an inland city in Northwestern China, located at the southern edge of the
Loess Plateau which is also an active source of mineral dust (Cao et al., 2008; Jeong, 2020;
Haugvaldstad et al., 2024), and the aging extent of mineral dust at Xi'an was found to be quite
limited (Wang et al., 2014; Wu et al., 2017). As Asian dust is transported eastward, it passes
over the North China Plain where anthropogenic emission is very high, and may become much
more aged when arriving at Qingdao, a coastal city of the Northwest Pacific (Li et al., 2014;
Pan et al., 2017). By comparing aerosol Al solubility at Xi'an and Qingdao, our work can
provide valuable insights into how and to which extent aging processes during long-range
transport can change aerosol Al solubility. Dust aerosol concentrations and meteorological
conditions vary remarkably at different seasons in Northern China; as a result, examining its
seasonal variations provides a good opportunity to understand the factors which regulate
aerosol Al solubility.
**2. Materials and methods**
**2.1 Sample collection**

Samples were collected at two cities (Xi'an and Qingdao) in Northern China at four

different seasons during 2021-2023 (Zhang et al., 2023; Chen et al., 2024), and further details
can be found in the supplement (Text S1 and Table S1). In brief, supermicron (>1 μm) and
submicron (<1 μm) particles were simultaneously collected using a two-stage aerosol sampler
(TH-150C, Tianhong Co., China) which was operated at 100 L/min, and the sampling duration
was typically 23.5 hours for each pair of aerosol samples. Whatman 41 cellulose filters were
used for aerosol collection in our work, and they were acid-washed before being used for
aerosol sampling to reduce background levels (Zhang et al., 2022). A total of 126 and 106 pairs
of aerosol samples were collected at Xi'an and Qingdao, respectively (Zhang et al., 2023; Chen
et al., 2024). After collection, all the aerosol samples were stored at -20°C for further analysis.
In addition to aerosol particles, we also sampled atmospheric acidic and alkaline gases
(mainly $NH_3$, HCl and $HNO_3$) at Qingdao, using a ChemComb 3500 Speciation Collection
Cartridge (Thermo Fisher Scientific, USA) at a flow rate of 10 L/min (Walters and Hastings,
2018; Fang et al., 2025). Gas sampling was carried out concurrently with aerosol sampling. In
brief, $NH_3$, $HNO_3$ and HCl were absorbed onto the inner walls of two tandem honeycomb
diffusion tubes coated with proper adsorbents, and then converted into $NH_4^+$, $NO_3^-$ and $Cl^-$.
After the sampling was completed, 20 mL ultrapure water was used to rinse each tube
immediately, and a PTFE membrane syringe filter (0.22 μm in pore size) was used to filter the
solution. The solution was then frozen at -20°C for further analysis.
**2.2 Sample analysis and aerosol acidity calculation**
Sample pretreatment and analysis were detailed in our previous work (Zhang et al., 2022),
and therefore are only briefly summarized here. The first half of a filter (and only one quarter
of a filter for supermicron particles) was shredded and digested in a Teflon jar using a
microwave digestion instrument. After digestion, the Teflon jar was filled with 1% $HNO_3$ (20
mL), and a PTFE membrane syringe filter (0.22 μm in pore size) was used to filter the solution;
subsequently, the solution was analyzed by inductively coupled plasma-mass spectrometry
(ICP-MS) to determine total concentrations of individual trace elements, including Al.
The other half of a filter was immersed in ultrapure water (20 mL) and stirred using an
orbital shaking for two hours; in the next step, the solution was filtered using a PTFE membrane
syringe filter (0.22 μm in pore size) and divided into two parts. The first solution was acidified
to contain 1% $HNO_3$ and subsequently analyzed by ICP-MS to determine the concentrations
of dissolved trace elements; the second solution was analyzed by ion chromatography (IC) to
quantify the concentration of water-soluble cations and anions.
The solutions obtained from honeycomb diffusion tubes (see Section 2.1 for more details)
were also analyzed using IC to determine the concentrations of gaseous $NH_3$, $HCl$ and $HNO_3$
in the atmosphere. ISORROPIA-II, a widely used aerosol thermodynamic model (Fountoukis
and Nenes, 2007), was employed in this work to calculate the acidity of supermicron and
submicron particles. It was operated in the forward mode, and aerosol particles were assumed
to remain metastable. Input parameters included concentrations of water-soluble ions in aerosol
particles and gaseous $NH_3$, $HCl$ and $HNO_3$, temperature and relative humidity (RH). Our
previous work found good agreement between measured and calculated $NH_3$ partitioning
coefficients at Qingdao (Fang et al., 2025), and as a result the method we used could well
estimate the acidity of supermicron and submicron particles.

## 3. Results

### 3.1 Seasonal variations of total and dissolved aerosol Al

### 3.1.1 Total aerosol Al

Figure 2 displays seasonal variations of total and dissolved aerosol Al at Xi'an and
Qingdao. At Xi'an (Figure 2a), total Al in supermicron particles showed highest concentrations
in spring and winter ($1.54\pm0.89$ and $1.91\pm0.93$ μg/m$^3$) and lowest concentrations in summer
($0.96\pm0.54$ μg/m$^3$); a similar seasonal pattern was observed for submicron particles, with total
Al concentrations being highest in spring and winter (4.29±3.70 and 2.92±1.47 μg/m³) and
lowest in summer (0.95±0.44 μg/m³). At Qingdao (Figure 2b), total Al concentrations in
supermicron particles were highest in spring (1.04±1.12 μg/m³) and lowest in summer and
autumn (0.33±0.18 and 0.31±0.12 μg/m³); similarly, for submicron particles, total Al
concentrations were also highest in spring (1.88±2.51 μg/m³) and lowest in summer and
autumn (0.35±0.22 and 0.65±0.82 μg/m³). For each season the median concentration of total
aerosol Al was usually higher in submicron particles than supermicron particles at both
locations (and there were some exceptions, as shown in Figures 1a and 1b). This is related to
size dependence of mineralogy and elemental compositions of mineral dust aerosol, which is
not well studied and deserves further investigation.

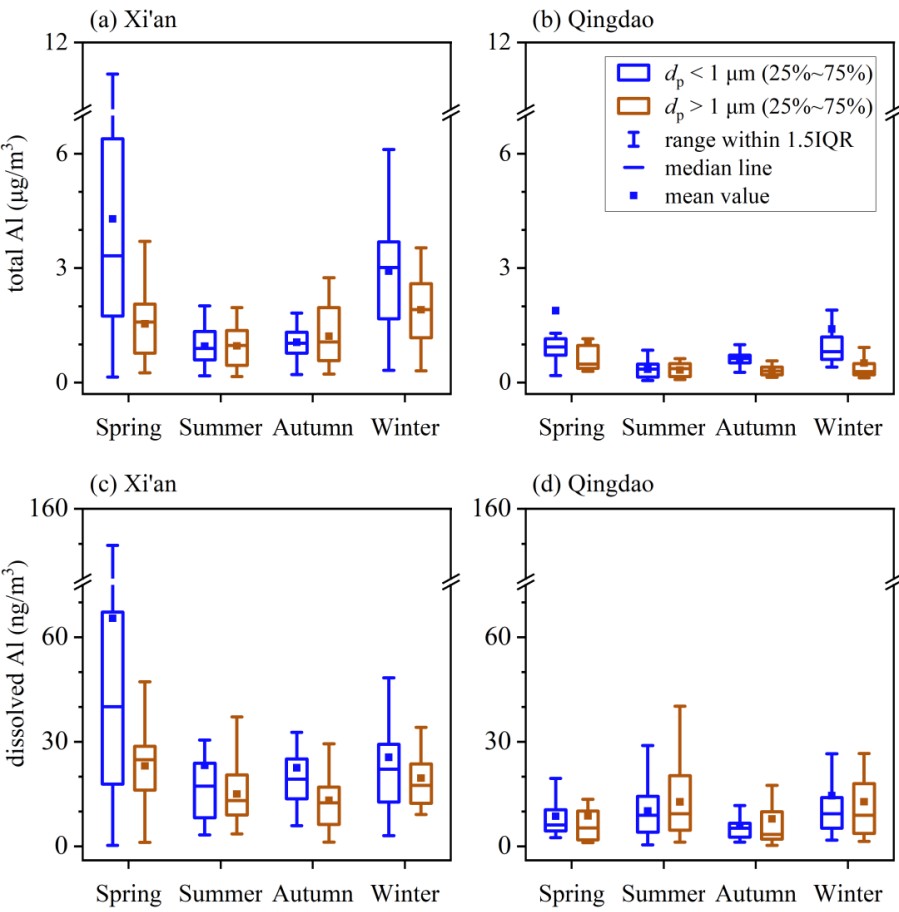


**Figure 2.** Seasonal variations of total and dissolved aerosol Al for submicron and supermicron
particles: (a) total Al at Xi'an; (b) total Al at Qingdao; (c) dissolved Al at Xi'an; (d) dissolved
Al at Qingdao.

Overall, total aerosol Al concentrations showed similar seasonal variations at Xi'an and
Qingdao, being highest in spring and lowest in summer. This was consistent with previous
studies carried out in other locations in East Asia, such as Zhengzhou (Wang et al., 2019),
Beijing (Zhang et al., 2013), Huaniao Island in the East China Sea (Guo et al., 2014), and Japan
(Sakata et al., 2023). In East Asia, mineral dust aerosol was emitted into the atmosphere mainly
in spring, leading to the increase in total aerosol Al concentrations. Lowest concentrations of
total aerosol Al were observed in summer because precipitation in Northern China mainly
occurred in summer, leading to enhanced wet deposition of aerosol particles (Cao and Cui,
2021). Furthermore, Qingdao was frequently affected by marine air masses in summer, and
this is also one reason why total aerosol Al concentrations were lower in summer than other
seasons. Total aerosol Al concentrations were higher in winter than summer and autumn at
Xi'an, and one major reason is that meteorological conditions favored the accumulation of
aerosol particles (including aerosol Al) during winter (Cao and Cui, 2021). Futhermore, besides
spring, Asian dust also occurs in winter (Cai et al., 2020; Wang et al., 2020), and a previous
study (Huang et al., 2014) suggested that the dust-related source, including local resuspended
dust, contributed 56% to $PM_{2.5}$ during a severe haze event at Xi'an.
As summarized in the supplement (Table S2), total aerosol Al concentrations exhibited
evident spatial variations in East Asia. As Asian dust was transported eastward to the North
Pacific, a clear decrease in aerosol Al concentrations was observed. Mineral dust was the
dominant source for aerosol Al, and therefore concentrations of aerosol Al were found to be
very high in desert regions. For example, total Al concentrations in TSP could reach 24 μg/m$^3$
over the Taklimakan Desert (Zhang et al., 2003). In our current study, annual average total Al
concentrations at Xi'an, an inland city close to the desert, were reported to be 1.42±0.86 and
2.28±2.35 μg/m$^3$ for supermicron and submicron particles, much lower than that observed over
the Taklimakan Desert. Further decrease in total Al concentrations was observed in coastal and
oceanic regions. For example, our work found that the annual average total Al concentrations
were 0.56±0.75 and 1.08±1.67 μg/m$^3$ for supermicron and submicron particles at Qingdao,
lower than those at Xi'an; total Al concentrations in TSP ranged from 0.17 to 1.72 μg/m$^3$ in
Hiroshima (Sakata et al., 2023), and further decreased to 1-56 ng/m$^3$ in Hawaii in the central
Pacific (Measures et al., 2010).
**3.1.2 Dissolved aerosol Al**
At Xi'an (Figure 2c), for supermicron particles, dissolved aerosol Al concentrations were
highest in spring (23.1±10.9 ng/m$^3$) and lowest in summer and autumn (15.0±8.7 and 13.2±8.6
ng/m$^3$); for submicron particles, dissolved Al concentrations were also highest in spring
(65.4±79.2 ng/m³) and lowest in summer and autumn (23.2±23.4 and 22.6±20.1 ng/m$^3$). Total
(Figure 2a) and dissolved aerosol Al (Figure 2c) showed similar seasonal patterns at Xi'an,
indicating that dissolved aerosol Al was mainly regulated by total aerosol Al.
As shown in Figure 2d, the average dissolved aerosol Al concentrations were 8.8±10.8,
12.8±11.1, 7.9±10.5 and 12.8±12.9 ng/m$^3$ for supermicron particles at Qingdao in spring,
summer, autumn, and winter, respectively, and 8.7±5.8, 10.2±8.2, 6.0±4.8 and 14.5±15.2 ng/m$^3$
for submicron particles. Dissolved aerosol Al concentrations were highest in summer and
winter and lowest in autumn for both supermicron and submicron particles. In contrast to Xi'an,
total and dissolved aerosol Al at Qingdao showed different seasonal patterns (Figures 2b and
2d); for example, total Al concentrations were lowest in summer at Qingdao when dissolved
Al concentrations were highest. This indicates that dissolved aerosol Al at Qingdao was not
only regulated by total aerosol Al but also affected by other factors such as atmospheric aging
processes.

Compared to Xi'an, dissolved Al concentrations at Qingdao were lower across all the four

seasons, mainly because total Al concentrations were much lower at Qingdao (Tables S3-S4 in
the supplement). As shown in Figure 2, similar seasonal patterns were observed at two
locations for total aerosol Al, but dissolved aerosol Al showed very different seasonality; this
suggests that seasonal patterns of aerosol Al solubility were different at Xi'an and Qingdao, as
presented in Section 3.2.
**3.2 Fractional solubility of aerosol Al**
**3.2.1 Seasonal variations of Al solubility**

Figure 3 displays aerosol Al solubility in different seasons at Xi'an and Qingdao. The

median solubilities of aerosol Al were determined to be 1.38%, 1.59%, 1.04% and 1.01% for
supermicron particles at Xi'an in spring, summer, autumn and winter, respectively, and 1.01%,
1.69%, 1.82% and 0.74% for submicron particles. Aerosol Al solubilities were generally low
for the four seasons at Xi'an, showing no apparent variation with seasons (Figure 3a). In
contrast, aerosol Al solubilities exhibited distinct seasonal variability at Qingdao (Figure 3b),
and the median Al solubilities were highest in summer (3.56% and 2.33%) and lowest in spring
(0.54% and 0.61%) for both supermicron and submicron particles.

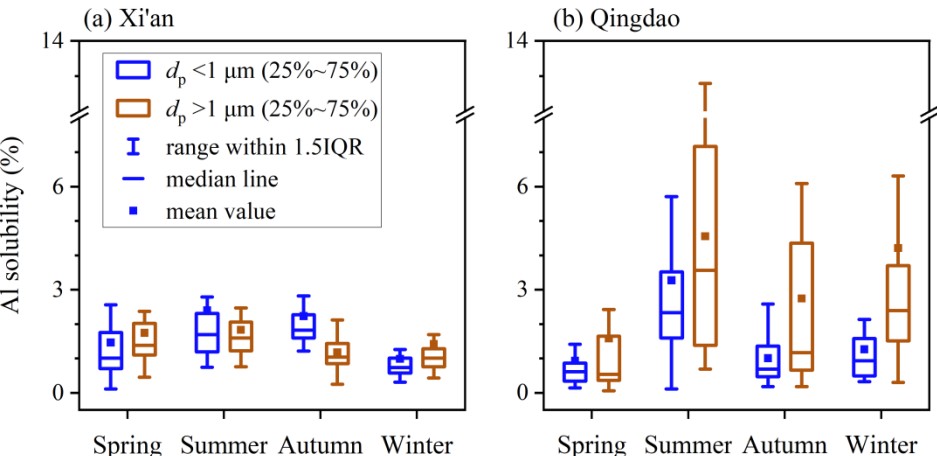

**Figure 3.** Seasonal variations of aerosol Al solubility for submicron and supermicron particles at (a) Xi'an and (b) Qingdao.

In three seasons (summer, autumn and winter), aerosol Al solubility at Qingdao was higher than that at Xi'an (Figure 3, Table S5). There are several important dust sources in Northwest China, being far from (up to a few thousand km) or close to Xi'an. More importantly, anthropogenic emission in Northwest China is much smaller than the North China Plain, and thus the aging extent of mineral dust transported to Xi'an was rather limited (Wang et al., 2014; Wu et al., 2017). On the contrary, Qingdao is much farther from deserts; consequently, after long-distance transport over the North China Plain where anthropogenic emission is very large, mineral dust aerosol which arrived at Qingdao was substantially aged (Trochkine et al., 2003; Takahashi et al., 2011; Jeong, 2020), thereby leading to enhanced dissolution of aerosol Al and thus the increase in Al solubility. Mineral dust from different desert regions and local suspended dust cannot explain higher Al aerosol solubility observed at Qingdao, as previous work showed that Al solubility was low for soil samples from different regions (Mulder et al., 1989; Duvall et al., 2008; Shi et al., 2011; Aghnatios et al., 2014; Li et al., 2022).

On the other hand, no obvious difference in aerosol Al solubility was observed between
Xi'an and Qingdao in spring, with median aerosol Al solubilities being <1.4% for supermicron
and submicron particles (Figure 3). This agrees with a previous study (Hsu et al., 2010) which
found that aerosol Al solubility was very low (average: ~0.7%) in spring even over the East
China Sea. Furthermore, similar to what we observed in spring at Xi'an and Qingdao, Al
solubility was found to be low (<1.5%) for surface soil particles (Mulder et al., 1989; Duvall
et al., 2008; Shi et al., 2011; Aghnatios et al., 2014; Li et al., 2022). Overall, our work implies
that in spring when Asian dust occurred most frequently, mineral dust particles arriving at
Qingdao after long-distance transport did not show substantial increase in Al solubility.
**3.2.2 Al solubility under different weather conditions**
We encountered four representative weather conditions (i.e. clean, dust, haze and fog days)
during our sampling at Xi'an and Qingdao, and investigated aerosol Al solubility under
different weather conditions (Figure 4, Tables S6-S7).
At Xi'an, no apparent difference in Al solubility was observed during clean, haze, and
dust days (Figure 4a, Table S6), with median values in the range of 1.01-1.47% for supermicron
particles and 0.72-1.86% for submicron particles. Al solubility was found to be <1.2% for three
mineral dust samples (Luochuan loess, Arizona test dust, and dust collected during a dust storm
in Xinjiang) (Li et al., 2022), and ranged from 0.47% to 1.42% for aerosol particles generated
using soil samples from Saharan desert (Shi et al., 2011). Compared to mineral dust in source
regions, Al solubility was not higher under different weather conditions at Xi'an. In addition,
although emission and accumulation of anthropogenic pollutants was greatly enhanced during
haze days at Xi'an (An et al., 2019; Cao and Cui, 2021), there was no obvious increase in

aerosol Al solubility, indicating that the effects of anthropogenic emissions on aerosol Al

solubility was limited at Xi'an. Therefore, one may conclude that aerosol Al solubility at Xi'an

was not different from initial Al solubility of mineral dust.

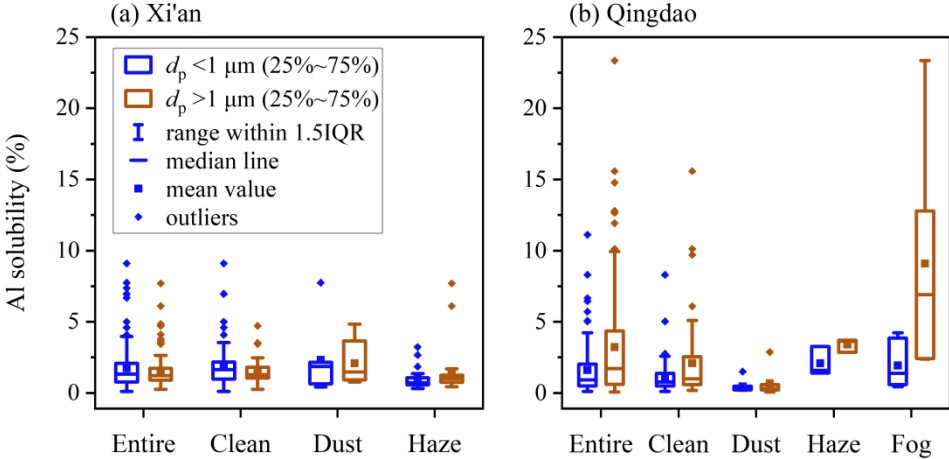

**Figure 4.** Aerosol Al solubility under different weather conditions for submicron and

supermicron particles: (a) Xi'an, (b) Qingdao.

Being different to Xi'an, aerosol Al solubility at Qingdao shows remarkable variations

under different weather conditions (Figure 4b, Table S7). Median Al solubilities were

determined to be 0.31% and 0.24% for supermicron and submicron particles during dust days,

lower than these on clean days (0.99% and 0.77%, respectively). This is probably because

higher wind speeds during dust events hindered the accumulation of atmospheric pollutants

and shortened the transport time to Qingdao, and thus limiting the aging of mineral dust aerosol.

This explanation is supported by a recent study (Zhang et al., 2024) which found that the aging

extent of dust particles in Japan was much lower during fast-moving dust events than slow-

moving dust events. Moreover, large amounts of alkaline components (such as carbonates)

which were emitted to the atmosphere during dust days neutralized acid species and therefore

inhibited acid-promoted dissolution of aerosol trace elements (Zhi et al., 2025). Our work
implies that during large dust events increase in aerosol Al solubility may be rather limited
when dust is transported to Qingdao; nevertheless, when dust is transported further eastward
to the open ocean, atmospheric chemical processing may substantially increase aerosol Al
solubility.
Figure 4b also suggests that aerosol Al solubilities were much higher during haze and fog
days at Qingdao, when compared to clean days. Highest Al solubilities were observed during
fog days, with median values being 6.90% for supermicron particles and 1.38% for submicron
particles, followed by haze days (3.64% and 1.58%, respectively). This is very likely due to
enhanced chemical processing during haze and fog periods (Shi et al., 2020; Shang et al., 2024),
and especially during fog days the large increase in RH cause huge increase in aerosol liquid
water, therefore greatly promoting aqueous reactions and Al dissolution. Acid and ligand
processing can both enhance aerosol Al solubility, although at present it is difficult to
disentangle their individual contributions.
In summary, aerosol Al solubility at Xi'an was low in general, and did not show much
variability in different seasons or under different weather conditions. Compared to Xi'an,
aerosol Al solubility was higher at Qingdao; furthermore, it was higher in the other three
seasons than in spring, and much higher for haze and fog days than dust days. These results
imply that atmospheric aging had little effect on aerosol Al solubility at Xi'an but could
remarkably increase aerosol Al solubility at Qingdao, as further elaborated in Section 4.

## 4. Discussion

As shown in Figure 5, our work observed the inverse dependence of aerosol Al solubility on total Al concentrations at both Xi'an and Qingdao, given by Eq. (1):

$$f_s(\text{Al}) = a \times [\text{Al}]^{-b} \qquad (1)$$

where $f_s(\text{Al})$ is aerosol Al solubility (%) and [Al] is total Al concentration (ng/m$^3$). Such relationship was also reported in some previous studies (Jickells et al., 2016; Shelley et al., 2018; Baker et al., 2020; Shelley et al., 2025). Baker and Jickells (2006) suggested that such inverse relationship was due to that larger particles have higher deposition velocities and lower Al solubility: aerosol Al concentrations decrease during transport in the atmosphere due to deposition, with deposition being faster for larger particles; as a result, aerosol particles will be enriched with smaller particles with higher Al solubility. However, Shi et al. (2011) found no substantial change in Al solubility with particle size for mineral dust samples, and therefore put the explanation proposed by Baker and Jickells (2006) into doubt.

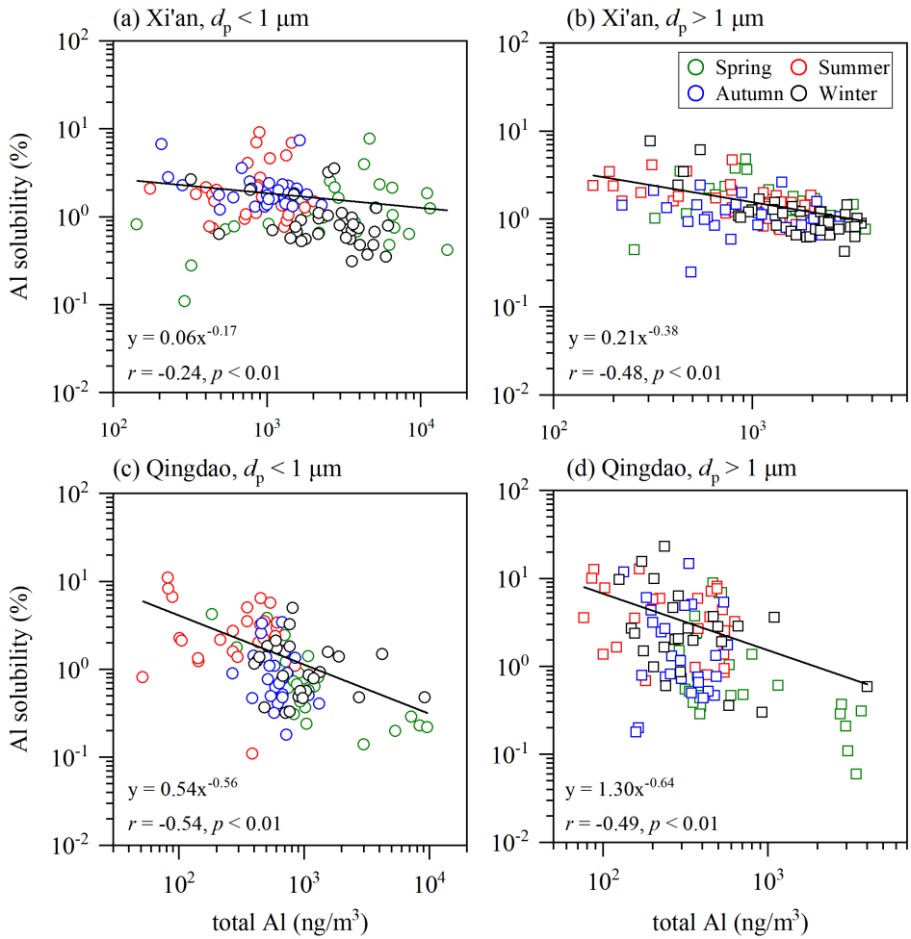

**Figure 5.** Aerosol Al solubility versus total aerosol Al concentrations: (a) submicron particles

at Xi'an, (b) supermicron particles at Xi'an, (c) submicron particles at Qingdao, (d)

supermicron particles at Qingdao.

Aerosol Fe solubility was also frequently observed to increase with the decrease in total

Fe concentrations (Sedwick et al., 2007; Mahowald et al., 2018; Meskhidze et al., 2019), and

one possible reason is the influence of anthropogenic aerosol Fe (Sholkovitz et al., 2009; Ito

and Shi, 2016) with higher solubility than mineral dust (Schroth et al., 2009; Fu et al., 2012;

Ito et al., 2021). Nevertheless, being different from aerosol Fe, aerosol Al stems

predominantly from mineral dust, with little contribution from anthropogenic sources;

furthermore, Al solubility was measured to be 0.4±0.6% for coal fly ash (Li et al., 2022), an
important type of anthropogenic aerosols, not higher than that for mineral dust (0.8±0.4%).
Therefore, we suggest that anthropogenic emission may not be able to explain the inverse
dependence of Al aerosol solubility on total Al concentrations.
We argue that chemical processing in the atmosphere can very well explain such inverse
dependence. Total aerosol Al concentrations decrease with transport due to deposition, while
reactions with acidic gases (such as $SO_2$ and $NO_x$) can enhance the dissolution of aerosol Al
(Jickells et al., 2016). Figure 5 shows that the inverse dependence of Al solubility on total Al
concentration was more pronounced at Qingdao, with the slopes (*b* values) much larger than
those obtained at Xi'an. This is because compared to Xi'an, Qingdao is more distant from
deserts and therefore dust aerosol is expected to be more aged at Qingdao. It also further
supports the vital role chemical aging plays in regulating aerosol Al solubility,
**4.1 Effects of acid processing and the role of RH**
**4.1.1 Effects of acid processing**
Laboratory experiments found that the amount of Al dissolved from minerals would
increase with the decrease in solution pH (Amram and Ganor, 2005; Bibi et al., 2011, 2014;
Cappelli et al., 2018), and some field measurements also suggested that acid processing in the
atmosphere could lead to large increase in aerosol Al solubility (Measures et al., 2010; Sakata
et al., 2023). In this work, we examined the relationship between aerosol Al solubility and the
relative abundance of acidic species ([sulfate]/[Al] and [nitrate]/[Al]) at Xi'an and Qingdao. It
should be noted that non-sea-salts sulfate (Virkkula et al., 2006), instead of sulfate, was used
at Qingdao because it is a coastal city and heavily impacted by sea spray aerosol.
At Xi'an, overall aerosol Al solubility showed no significant correlation with [sulfate]/[Al]
or [nitrate]/[Al] for either supermicron or submicron particles ($r < 0.4$, Figure 6 and S1),
indicating that acid processing did not enhance aerosol Al solubility. Enhancement of aerosol
trace element solubility by acid processing requires internal mixing of acid species with mineral
dust particles (Baker and Croot, 2010). Previous studies suggested that mineral dust particles
observed at Xi'an which is close to deserts largely remained externally mixed with acid species
(Wang et al., 2014; Wu et al., 2017), and thus aerosol Al solubility was not apparently enhanced
by acid processing at Xi'an.

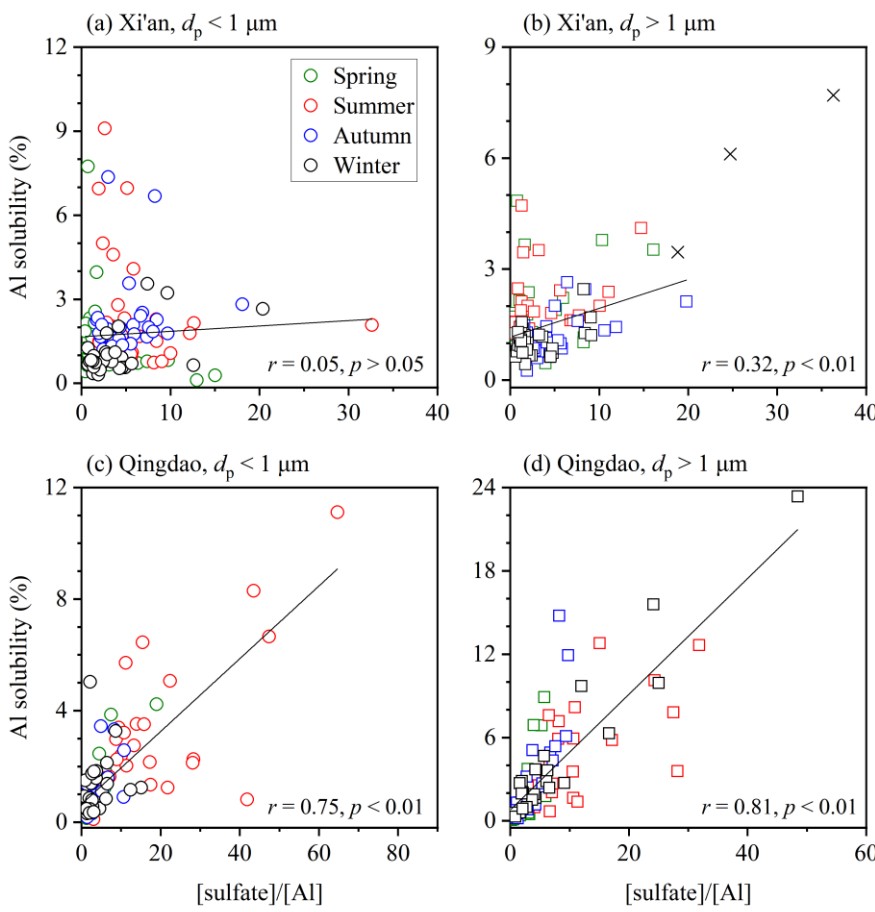


**Figure 6.** Aerosol Al solubility versus [sulfate]/[Al]: (a) submicron particles at Xi'an, (b)
supermicron particles at Xi'an, (c) submicron particles at Qingdao, (d) supermicron particles
at Qingdao (the *r* value changed from 0.81 to 0.74 if the data point with the highest Al solubility
was excluded). Data represented by crosses are not included in fitting.

On the contrary, Figure 6 shows that aerosol Al solubility at Qingdao was well correlated
with [sulfate]/[Al] ($r > 0.7$, $p < 0.01$), implying that acidic species were internally mixed with
mineral dust particles and thus acid-promoted dissolution significantly enhanced Al solubility.
We also found that correlations of Al solubility with [sulfate]/[Al] was better than those with
[nitrate]/[Al] (Figures 6 and S1, Table S8), in line with a previous study (Sakata et al., 2023)
which found aerosol Al solubility at Hiroshima, southern Japan, to be correlated with
[sulfate]/[Al] but not with [nitrate]/[Al]. This may imply that chemical processing by sulfate
was more important than nitrate for Al solubility enhancement via acid processing, likely
because aluminosilicate dust particles tend to react preferentially with $SO_2$ and $H_2SO_4$ while
nitrogen oxides react mainly with carbonate particles (Sullivan et al., 2007; Fitzgerald et al.,
2015). Furthermore, our work reveals better correlations between Al solubility and [sulfate]/[Al]
for supermicron particles than submicron particles (Figure 6), indicating that the effect of acid
processing on Al solubility was more important in supermicron particles.
**4.1.2 The role of RH**
Relative humidity (RH) is a vital factor influencing liquid water contents and phase state
of aerosol particles and thus their secondary chemistry. When RH increased >60%, the phase
state of aerosol particles in Northern China changed from semisolid to liquid (Liu et al., 2017;
Sun et al., 2018; Song et al., 2022), leading to large increase in aerosol liquid water content and
thereby potentially affecting aerosol Al solubility.
We observed no apparent variation of aerosol Al solubility with RH at Xi'an (Figure 7a).
When RH was <60%, median Al solubilities for supermicron and submicron particles were
1.22% and 1.14%, respectively; when RH increased >90%, the median Al solubilities were
determined to be 1.82% and 0.82%, showing no obvious increase when compared to those at
<60% RH. This again may imply that chemical processing had very limited impact on aerosol
Al solubility at Xi'an, as mineral dust particles mostly remained externally mixed with
secondary species and their aging extent was very limited (Wang et al., 2014; Wu et al., 2017).

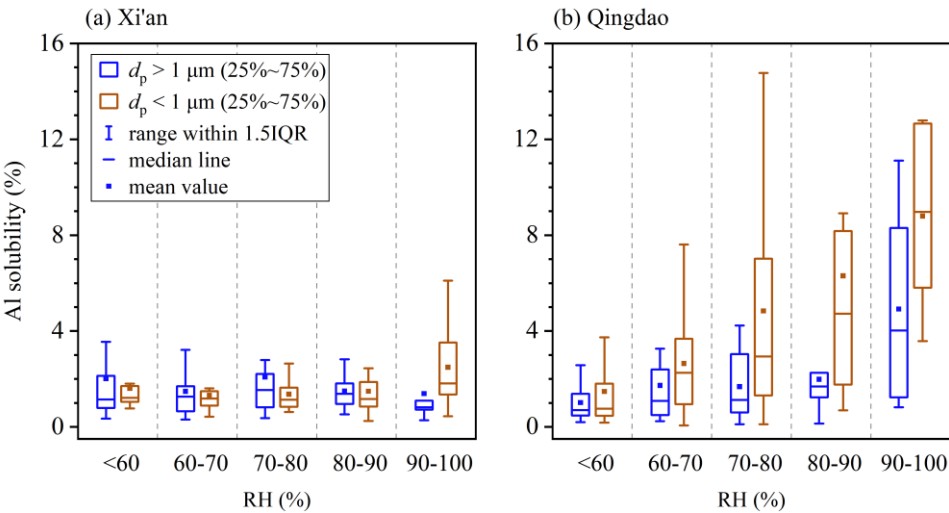


**Figure 7.** Aerosol Al solubility at different relative humidity (RH) for submicron and
supermicron particles: (a) Xi'an, (b) Qingdao.

In contrast, RH played an important role in regulating aerosol Al solubility at Qingdao,
because mineral dust particles observed at Qingdao had been transported through the North
China Plain and were substantially aged. As shown in Figure 7b, for supermicron particles, the
median Al solubility was only 0.76% at <60% RH, and gradually increased to 4.73% at 80-90%
RH, and abruptly increased to 8.87% at >90% RH. For submicron particles, median Al
solubility was <1% at <60% RH, and further increase in RH to 80-90% did not lead to large
changes in Al solubility; nevertheless, when RH exceeded 90%, the median Al solubility was
remarkably increased to 4.02%, much higher than those observed when RH was < 90%.
**4.2 Effects of aerosol acidity on aerosol Al solubility at Qingdao**

Figure 8 shows the dependence of aerosol Al solubility on aerosol acidity (represented by

pH) at Qingdao (we did not measure $NH_3$ at Xi'an and thus could not estimate the aerosol
acidity in a reliable manner). For supermicron particles, the median Al solubility was only 0.99%
when aerosol pH was >4.0, and gradually increased to 10.24% as aerosol pH was decreased to
<2.5. For submicron particles, the median Al solubility was only 0.69% when pH was >4.0,
increased slightly with the decrease in pH when pH was in the range of 2.5-4.0, and then
increased greatly to 6.09% when pH was decreased to <2.5. In addition, aerosol acidity at
Qingdao was highest in summer and lowest in spring (Chen et al., 2024), consistent with the
seasonal variation of aerosol Al solubility, further supporting the importance of aerosol acidity
in regulating Al solubility.

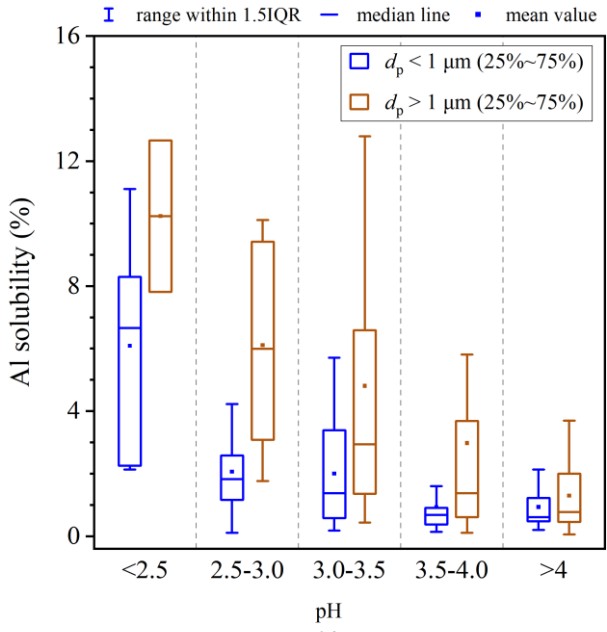


**Figure 8.** Aerosol Al solubility corresponding to different aerosol acidity for submicron and
supermicron particles in Qingdao.

As shown in Figure S2, aerosol Al solubility was generally <2% when aerosol acidity was
low (pH > 4.0), and higher Al solubility (>2%) was usually observed for samples with high RH
and high acidity (pH < 4.0), again underscoring the roles of aerosol acidity (and RH). However,
some samples exhibited low Al solubility although the corresponding RH and aerosol acidity
were both higher, and such phenomenon was more pronounced for submicron particles. This
is very likely linked with aerosol mixing state (Riemer et al., 2019). Aerosol Al solubility and
acidity used in our work are both the average properties of an aerosol sample which contains
numerous particles, while in reality the two properties will have large particle-to-particle
variations. For a given aerosol sample, it can happen that particles with high acidity may
contain very little Al while particles with low acidity are enriched in Al; in this case, high
acidity do not promote Al solubility for this sample. Single particle analysis which provides
mixing state information can give further insights. We also note that samples with low Al
solubility but high RH and high acidity were mostly found in clean days, perhaps due to the
influence of local resuspended dust for which chemical aging was very limited.
**4.3 Size-dependence of aerosol Al solubility**
At Xi'an, no obvious difference in aerosol Al solubility was found between supermicron
and submicron particles across all the four seasons (Figure 3a). This is because the aging extent
of dust particles was rather limited at Xi'an (Wang et al., 2014; Wu et al., 2017) and Al
solubility does not vary with particle size for unaged dust particles (Shi et al., 2011). At
Qingdao, aerosol Al solubility showed no obvious difference between supermicron and
submicron particles in spring, because the aging extent of dust arriving at Qingdao was also
limited in spring when Asian dust occurred most frequently. However, in the other three
seasons, Al solubility was higher for supermicron particles than submicron particles at Qingdao,
and the ratios of median Al solubility in supermicron particles to that in submicron particles
were found to be 1.53, 1.70 and 2.57 in summer, autumn and winter, respectively. Similar to
our observation at Qingdao, Li et al. (2017) found that aerosol Al solubility was much higher
for TSP (14-28%) than $PM_{2.5}$ (2-23%) at the summit of Mount Heng, southern China.

On the other hand, a few other studies (Baker et al., 2020; Hsieh et al., 2023; Sakata et al.,

2023; Yang et al., 2023) found that aerosol Al solubility was higher in fine particles than coarse
particles. For example, aerosol Al solubility was found to increase with the decrease in particle
size over the tropical eastern Atlantic (Baker et al., 2020), being ~10.31% for particles in the
size of 0.36-0.61 μm and 0.43-4.53% for particles above 0.61 μm. At Hiroshima, southern
Japan, aerosol Al solubility was reported to be 8.82±6.48% for fine particles (<1.3 μm), more
than two times larger than that (3.25±3.41%) for coarse particles (>1.3 μm) (Sakata et al., 2023).
Baker and Jickells (2006) suggested that this is because fine particles have larger surface-to-
volume ratios and thus facilitate Al dissolution via acid processing. Hsieh et al. (2023) found
aerosol Al solubility to be 38% for fine particles (0.57-1.0 μm) but only 0.37% for coarse
particles (>7.3 μm) over the East China Sea, and suggested that the observed size-dependence
could be explained by the enrichment of anthropogenic Al (which has higher solubility than
dust Al) in fine particles. However, aerosol Al originates predominantly from mineral dust,
with little contribution from anthropogenic sources (Taylor and McLennan, 1985; Mahowald
et al., 2018), and fractional solubility of anthropogenic Al was not necessarily higher than
mineral dust (Li et al., 2022).
As discussed above, there is not clear yet how and why aerosol Al solubility varies with
particle size. Such discrepancy is at least partly because different leaching protocols were used
in previous studies to extract dissolved aerosol Al and thereby Al solubility obtained in
different studies was not directly comparable (Meskhidze et al., 2019; Li et al., 2023; Li et al.,
2024). Furthermore, mechanistic insights can be obtained by laboratory experiments which
examine the size dependence of the solubility and dissolution kinetics of Al for mineral dust
particles under atmospherically relevant conditions.

## 5. Conclusions and atmospheric implications

Deposition of mineral dust aerosol is a major external source of several nutrient and toxic
elements for surface water in open oceans, and thus have large impacts on marine
biogeochemistry; however, previous studies which estimated dust deposition flux into the
oceans reveals large discrepancies. Aerosol Al solubility, which is a critical parameter in using
dissolved Al concentrations in surface seawater as a tracer to constrain dust deposition flux,
remains poorly understood. In this work, we investigated seasonal variations of aerosol Al
solubility for supermicron (>1 μm) and submicron (<1 μm) aerosol particles at Xi'an and
Qingdao, both located in Northern China, in attempt to elucidate the processes and mechanisms
which govern the variation of aerosol Al solubility in the atmosphere.
At Xi'an, aerosol Al solubility was low in general for both supermicron and submicron
particles, showing no obvious variability in different seasons or under different weather
conditions. This implies that chemical processing did not substantially enhance aerosol Al
solubility at Xi'an, as it is an inland city close to major deserts in Northwestern China and thus
the aging extent of mineral dust particles arriving at Xi'an was quite limited. Compared to
Xi'an, aerosol Al solubility was higher at Qingdao, a coastal city in Northern China;
furthermore, Al solubility was higher in the other three seasons than in spring, and much higher
for haze- and especially fog-impacted days than dust days. This indicates that chemical
processing substantially increased aerosol Al solubility at Qingdao.

Aerosol Al solubility at Xi'an showed no significant correlation with relative abundance

of sulfate or nitrate, and did not vary apparently with RH; in contrast, Al solubility at Qingdao
was well correlated with relative abundance of sulfate and nitrate, and increased with RH. This
further supports that chemical processing had little impact on aerosol Al solubility at Xi'an
(because the aging extent of mineral dust aerosol at Xi'an is very limited) but remarkably
increased aerosol Al solubility at Qingdao (because mineral dust particles transported to
Qingdao were substantially aged). Moreover, for both supermicron and submicron particles,
Al solubility at Qingdao was found to increase with aerosol acidity (in addition to RH),
underscoring the vital role of aerosol liquid water and acidity in enhancing Al dissolution via
chemical aging.

Our comprehensive investigation of aerosol Al solubility at two locations in Northern

China suggests that atmospheric chemical processing dictates aerosol Al solubility. As a result,
aerosol Al solubility is expected to spatially variable, depending on the extent of chemical
processing. For example, we found that aerosol Al solubility is higher at Qingdao than Xi'an
in general, and expect it to increase further as mineral dust aerosol is further transported
eastward to the Pacific. Although our measurements were only conducted at two sites, our work
provides important insights into processes driving spatiotemporal variability of aerosol Al
solubility, and such understanding can aid us to develop aerosol Al solubility parameterizations.

**Author contribution.**
**TZ:** Formal analysis, Investigation, Writing - Original Draft, Writing - Review & Editing;
**YC:** Formal analysis, Investigation, Writing - Original Draft; **HZ:** Investigation; **Lei Liu:**
Writing - Review & Editing; **CH:** Investigation; **ZF:** Investigation; **YZ:** Investigation; **FW:**
Resources; **Lan Luo:** Resources; **GZ:** Writing - Review & Editing; **XW:** Resources; **MT:**
Conceptualization, Formal analysis, Supervision; Writing - Original Draft, Writing - Review
& Editing.
**Competing interests.**
The authors declare that they have no conflict of interest.
**Acknowledgement.**
We would like to thank colleagues at Shandong University, Shaanxi University of Science
and Technology, and Institute of Earth Environment, Chinese Academy of Sciences for their
support during field measurements.
**Financial support.**
This work was sponsored by National Natural Science Foundation of China (42277088,
42407149 and 22361162668), Guangzhou Bureau of Science and Technology
(2024A04J6533), International Partnership Program of Chinese Academy of Sciences
(164GJHZ2024011FN), Guangdong Basic and Applied Basic Research Fund Committee
(2023A1515012010), and Guangdong Foundation for Program of Science and Technology
Research (2023B1212060049).

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
