# Peer review of "Atmospheric chemical processing dictates aerosol aluminum solubility: insights from"

_EGUsphere, 2025_

## Author Comment (AC1)

Comments by referees are in blue.
Our replies are in black.
Changes to the manuscript are highlighted in red both here and in the revised manuscript.

**Reply to referee #2**

The paper systematically investigates the seasonal variations in aerosol aluminum solubility in two northern Chinese cities (Xi'an and Qingdao), revealing the critical influence of atmospheric chemical processes on aluminum solubility. The research topic holds certain scientific significance, as the study of aerosol aluminum solubility is expected to provide key parameters for accurately estimating mineral dust deposition fluxes in the context of marine biogeochemistry and global climate change. However, the paper devotes a significant portion of its content to presenting test data without in-depth analysis or detailed interpretation, resulting in insufficient depth and scientific value. Many conclusions are drawn at a speculative level, lacking solid evidence to substantiate them, which undermines the credibility of the article. The core conclusions of the paper currently carry considerable uncertainty and require further refinement; they should not be hastily drawn. Extensive revisions are recommended, and my suggestions and comments are as follows.

**Reply:** We would like to thank ref #2 for reviewing our manuscript and recommending it for publication after major revision.

We do not quite agree with the comment that we present data without in-depth analysis or interpretation. Our original manuscript uses ~7 pages (page 8-15) to present our result (Section 3), and ~10 pages (page 15-24) to analyze, discuss and interpret our results (Section 4). Due precisely to our in-depth data analysis and discussion, we could explain the difference in aerosol Al solubility at two locations and come to the conclusion that atmospheric chemical processing dictates aerosol aluminum solubility. There may be other unknown mechanisms which can explain our data, but the concolusion we have reached (i.e. atmospheric chemical processing dictates aerosol aluminum solubility) are consistant with our results and scientifically robust.

Nevertheless, we understand that there is always plenty of room for improvement. We highly respect and appreciate all the comments ref #2 raised, and have revised our manuscript accordingly; when we do not quite agree with ref#2, we have provided proper explanation. Please find more details below.

1. The abstract is overly vague and generalized. For instance, it should explicitly summarize the seasonal variation patterns of aluminum solubility in the two regions, the correlation with the relative abundance of sulfate and nitrate, and how their dependence on relative humidity (RH) differs between the two locations, rather than merely stating that they are "different."

**Reply:** Indeed we would like to provide further details about aerosol Al slolubility in the abstract. However, the abstract cannot exceed 250 words, as required by the journal. As a result, we cannot provide these details in the abstract; instead, we present them in the conclusion.

2. It is unclear why the authors claim that spatial differences in aerosol aluminum solubility must be fully considered when constraining oceanic dust deposition using dissolved aluminum concentrations in surface seawater. Although the solubility of aluminum in dust differs between Xi'an and Qingdao, and the authors attribute this to varying degrees of aging, dust transported from the same source region to the same oceanic area should undergo the same aging process, resulting in consistent aluminum solubility. It is unclear how the authors arrived at this conclusion.

**Reply:** We respectlt disagree with ref #2. First of all, over the oceans aerosol Al solubility can show spatial variation; furthermore, for a given region over the coean, aerosol Al solubility can vary great temporal variation, as dust particles arriving at a same point but at different times may have undergone different aging processes.

In fact, as pointed out in our original manuscript (page 3, line 62-64), previous work found that aerosol Al solubility could vary over the oceans by more than one order of magnitude.

3. The Introduction fails to clearly focus on the current major controversies and sources of uncertainty regarding the range of aluminum solubility variations and their influencing mechanisms. Some studies suggest a significant correlation between aluminum solubility and acidic components, while others hold opposing views. The Introduction should more clearly summarize these conflicting findings and unresolved mechanisms, identifying the specific "gaps" or "contradictions" the current study aims to address, thereby strengthening the research motivation.

**Reply:** Ref #2 rasied a very good point. As suggested, in the revised manuscript (page 5) we have added a few senetences to summarize the key gaps in our understanding of aerosol Al solubility: "It can be concluded that although aerosol Al solubility in the atmosphere was explored by several previous studies, our understanding is still very limited. For example, it remains unclear why aerosol Al solubility shows large spatial and temporal variation. Some work suggested that atmospheric chemical aging could enhance aerosol Al solubility, but the mechanisms and key environmental factors have not been elucidated. Furthermore, the effects of particle size on aerosol Al solubility have not been well understood."

4. How were the interferences from locally resuspended dust aluminum in Xi'an and Qingdao excluded? How is it proven that the dust observed in Xi'an and Qingdao originates from the same source and differs only in aging?

**Reply:** Indeed local resuspended dust, in addition to desert dust, could contribute to aerosol Al at both locations. In response to this comment, we have made the following changes:

1) In the revised manuscript we have changed "desert dust" to "mineral dust" so that we do not exclude the possible contribution of local resuspended dust.

2) Mineral dust from different desert regions and local resuspended dust cannot explain higher Al aerosol solubility in Qingdao, as previous work showed that Al solubility was low for soil samples from different regions. In the revised manuscripy (page 13) we have added one sentence to provide further explanation: "Mineral dust from different desert regions and local suspended dust cannot explain higher Al aerosol solubility observed at Qingdao, as previous work showed that Al solubility was low for soil samples from different regions (Shi et al., 2011; Wuttig et al., 2013; Aghnatios et al., 2014; Li et al., 2022; Hsieh et al., 2023)."

5. The study claims that its findings can be generalized to the "North Pacific dust pathway" or even the "global dust-ocean interface." However, the current design is based on only two sampling sites (one inland and one coastal), lacking gradient observations (e.g., multi-point trajectory analysis) or broad representativeness. The observed differences may be dominated by the unique characteristics of the sampling sites themselves (e.g., urban pollution, local humidity). How can such significant uncertainty be explained?

**Reply:** We think that this comment is related to the last paragraph in our manuscript where we discuss the implication of our work.

We found that compared to Xi'an (an inland site), aerosol Al solubility at Qingdao (a coastal site) was much higher, and we attributed this to chemical processing; as a result, it is justified to expect that aerosol Al solubility will further increase when mineral dust aerosol is transported to the Pacific. We also found that aerosol Al solubility at Qingdao showed

temperail variations, as the extent of aging also varied with time; as a result, at a given location over the oceans, aerosol Al solubility may also vary with time. Therefore, it is very justified to state that "when leveraging dissolved Al concentrations in surface seawater as a tracer to estimate deposition flux of mineral dust aerosol into open oceans, considering the spatial distribution of aerosol Al solubility, instead of using a uniform value on the global scale, can help us better constrain the oceanic deposition flux of mineral dust."

6. Why is aluminum concentration highest in winter? Mineral dust is not commonly observed in Xi'an during winter. Does this indicate that the source of aluminum in Xi'an is not mineral dust?

**Reply:** In fact many studies showed that mineral dust is a major component in aerosol particles at Xi'an. The major reason why aerosol Al concentrations was higher in winter than summer and autumn is that meteorological conditions in winter favored accumulation of aerosol particles. In the revised manuscript (page 10) we have added one sentence for further explanation: "Total aerosol Al concentrations were higher in winter than summer and autumn at Xi'an, and one major reason is that meteorological conditions favored the accumulation of aerosol particles (including aerosol Al) during winter (Cao and Cui, 2021)."

7. Lines 170-173: All the sites the authors compare are island observations, which are not strongly comparable to Xi'an. There are numerous observational results from inland China—why are these not mentioned for comparison?

**Reply:** Ref #2 rasied a good point. Total aerosol Al showed similar seasonal variations at other inland sites in North China (such as Zhengzhou and Beijing). As suggested, in the revised manuscript (page 10) we have included two studies carried out at Zhengzhou and Beijing, and deleted the two studies carried out at Hong Kong and Taiwan: "This was consistent with previous studies carried out in other locations in East Asia, such as Zhengzhou (Wang et al., 2019), Beijing (Zhang et al., 2013), Huaniao Island in the East China Sea (Guo et al., 2014), and Japan (Sakata et al., 2023)."

8. Lines 184-193: The comparative data presented here merely show that aluminum concentrations are higher at sites closer to dust source regions—a conclusion that is obvious and lacks significant scientific value. Could the authors supplement the discussion with differences in aluminum content (μg/g) in dust aerosols at sites at varying distances from dust sources? Analyzing changes in aluminum content during transport and their underlying mechanisms would be more scientifically valuable. The same applies to the analysis of soluble aluminum. Readers would prefer to see variations in aluminum content rather than just absolute concentration changes related to distance from dust sources.

**Reply:** It can be a good alternative to discuss change in aluminmun content at different sites; however, such data is not available as most of previous and our studies only report mass concentrations of aerosol Al. It can be expected that the increase in the transport distance will lead to decrease in Al content, since Al concentrations will decrease gradually while other aerosol components may increase.

9. The study presents aluminum concentrations in supermicron and submicron particles but merely displays the data without explaining its scientific significance. Why do aluminum concentrations differ between particle sizes? What mechanisms underlie these differences? How do the seasonal variation characteristics of aluminum content differ between particle sizes, and what causes these differences? The authors' data analysis needs strengthening; it should not be limited to simple data presentation.

**Reply:** In the revised manuscript (page 9) we have added the following sentences to describe and explain the general feature of size dependence of aerosol Al: "For each season the median concentration of total aerosol Al was usually higher in submicron particles than supermicron particles at both locations (and there were some exceptions, as shown in Figures 1a and 1b). This is related to size dependence of mineralogy and elemental compositions of mineral dust aerosol, which is not well studied and deserves further investigation." We also would like to point out that the focus of our manuscript is to understand aerosol Al solubility (as discussed in Section 4) while Section 3 is used to present relevant results .

10. Lines 220-224: What drives the seasonal variation in aluminum solubility? Why do differences in aluminum solubility exist between particle sizes? How do seasonal variations in aluminum solubility differ between particle sizes, and why? If there are no differences between particle sizes, then studying size-dependent features is unnecessary.

**Reply:** In fact, in our original manuscript we have discussed what drives the seasonal patterns of aerosol Al solubility at Xi'an and Qingdao (Section 4.2), and also discussed size dependence of aerosol Al solubility (Section 4.3). Ref #2 is kindly referred to relevant sections for further details.

11. Line 233: Here, the difference in aluminum solubility between Xi'an and Qingdao is attributed to transport distance. What evidence supports this claim, or is it merely speculation? How large is the uncertainty of this speculation, and how can it be validated?

12. The authors attribute the differences in aluminum solubility between the two cities to aging during transport. However, it should be noted that the distance from Xi'an to the Taklamakan Desert exceeds 3,000 km, while the distance from Xi'an to Qingdao is about 1,000 km. In other words, transport from Xi'an to Qingdao increases aging time by only about 30%, which is not a substantial difference. Without solid evidence proving the significance of this 1,000 km aging process, the core conclusion is difficult to accept. At present, this conclusion appears to be speculative.

16. Line 258: If the aluminum solubility in Xi'an's dust is very close to that at the source region, how can the impact of aging over nearly 3,000 km of transport on aluminum solubility be explained? Conversely, why does the 1,000 km transport from Xi'an to Qingdao have such a pronounced effect on aluminum solubility?

**Reply:** These three comments (No. 11, 12 and 16) are closely related, and therefore are addressed together. Aerosol Al solubility at Xi'an was not different from for dust samples collected over deserts, while was much higher at Qingdao. As we further discuss in Section 4, we suggest that all the features related to aerosol Al solubility can be only explained by the extent of chemical aging.

In addition to the Taklamakan Desert which is ~3000 km from Xi'an, there are several important dust sources in Northwest China which are quite close to Xi'an, such as China Loess Planteau. More importantly, anthropogenic emission over dust source regions in Northwest China is much smaller than North China Plain, and therefore mineral dust particles arriving at Xi'an are not much aged, as supported by previous work. In the revised manuscript (page 13) we have made the following changes to provide further explanation: "In three seasons (summer, autumn and winter), aerosol Al solubility at Qingdao was higher than that at Xi'an (Figure 3, Table S5). There are several important dust sources in Northwest China, being far from (up to a few thousand km) or close to Xi'an. More importantly, anthropogenic emission in Northwest China is much smaller than the North China Plain, and thus the aging extent of mineral dust transported to Xi'an was rather limited (Wang et al., 2014; Wu et al., 2017). On the contrary,

Qingdao is much farther from deserts; consequently, after long-distance transport over the North China Plain where anthropogenic emission is very large, mineral dust aerosol which arrived at Qingdao was substantially aged (Trochkine et al., 2003; Takahashi et al., 2011; Jeong, 2020), thereby leading to enhanced dissolution of aerosol Al and thus the increase in Al solubility."

13. Line 239: Why is the difference in aluminum solubility between the two cities minimal in spring? Spring is precisely the season when dust events are most significant in inland China. If aluminum solubility does not exhibit differences during this critical dust period, does this suggest that the tested aluminum solubility differences are not attributable to dust-derived aluminum?

**Reply:** As we further discuss in the original manuscript (line 268-278), aerosol Al solubility in spring was not different from dust particles collected over desert regions at either Xi'an or Qingdao. This is because higher wind speeds during dust events hindered the accumulation of atmospheric pollutants and shortened the transport time, limiting the aging of mineraldust aerosol.

Initial Al solubility was very low (and essentially identical) for mineral dust particles emitted from different regions. The difference in aerosol Al solubility at Xi'an and Qingdao was caused by atmospheric chemical aging, and this is the central message our work wants to deliver.

14. On the other hand, the impact of aluminum deposition on marine ecosystems should be evaluated from the perspective of dust load weighting. If, as the study shows, aluminum solubility is lowest during spring—when dust loads are highest—then the vast majority of annual dust-derived aluminum has poor solubility, calling into question its contribution to marine ecosystems. Moreover, does this imply that aluminum solubility in Xi'an and Qingdao is largely similar for most of the year, with differences only appearing in the less dusty seasons (summer, autumn, and winter), thus contributing minimally to annual totals? The authors should scientifically assess these differences in light of total dust transport.

**Reply:** I agree with ref #2 that one may conclude from our work that aerosol Al solubility is low in spring when dust loads are highest. However, it is beyond the scope of our manuscript to discuss the impact of dust deposition on marin ecosystems. The key meassage we would like to deliver is that atmospheric chemical aging dictates aerosol aluminum solubility and can explain its spatial and temporal variations.

15. Line 252: What are the concentrations of dust aerosols under different weather conditions? On clean or haze days, there is likely little dust transported from deserts, with local resuspended dust dominating. If aluminum solubility is similar under these conditions, does this suggest that solubility differences between desert dust and local resuspended dust in Xi'an are negligible, making it unreliable as a source indicator?

17. Line 283: How is it proven that dust sources are consistent across haze days, fog days, and clean days? How is it confirmed that aluminum solubility at the origin is identical and that the observed differences arise primarily from varying atmospheric chemical processes?

**Reply:** The two comments above (No. 15 and 17) are closely related, and thus are addressed together. In our original supplement (Tables S6 and S6) we have provided total and dissoleved aerosol concentrations under different conditions at Xi'an and Qingdao. Total aerosol Al concentrations were much higher for dust days. Because our manuscript is focused on aerosol Al solubility, we choose not to discuss in specific total Al aerosol concentrations under different weather conditions.

Indeed we cannot exclude the contribution of local resuspended dust. As a result, in the revised manuscript we have changed "desert dust" to "mineral dust". It is true that mineral dust aerosol at a given location can orginate from different source regions; however, original mineral dust, even from different regions, all shows very low Al solubility, as documented by a number of previous work. In the revised manuscripy (page 13) we have added one sentence to provide additional justication: "Mineral dust from different desert regions and local suspended dust cannot explain higher Al aerosol solubility observed at Qingdao, as previous work showed that Al solubility was low for soil samples from different regions (Shi et al., 2011; Wuttig et al., 2013; Aghnatios et al., 2014; Li et al., 2022; Hsieh et al., 2023)."

18. Line 285: The authors vaguely speculate that liquid-phase reactions enhance aluminum solubility. First, dust is a hydrophobic aerosol, and even on haze days, aerosol water content does not increase significantly. Can the authors provide data on how much aerosol liquid water content actually increased during their observations? Furthermore, what specific liquid-phase reactions promote aluminum dissolution? What triggers these reactions? Why do such reactions not occur on haze days in Xi'an?

**Reply:** Unaged mineral dust particles are largely hydrophobic, but they can still take up some water at high RH. More important, aging will increase the hygroscopicity of mineral dust. This exactly explains why aerosol Al solubility increases with RH at Qingdao (where dust particles are aged) but does not vary with RH at Xi'an (where dust particles are still not aged).

Based on previous literature and what we know for aqueous chemistry of Al, we suggest that acid and ligand processing can both enhance aerosol solubility, but at present it is difficult to disentangle their individual contributions. In the revised manuscript (page 16) we have added one sentence for further discussion: "Acid and ligand processing can both enhance aerosol Al solubility, although at present it is difficult to disentangle their individual contributions."

19. Based on the authors' analysis, the observed increase in aluminum solubility in Qingdao is more likely due to local atmospheric chemical processes (if their speculation is correct) rather than aging during transport. On haze days in Qingdao, it is unlikely that the entire dust transport pathway experiences haze conditions; instead, these are more likely dust days, representing cases with minimal aging and the least increase in aluminum solubility. Dust days are the most critical cases for annual dust transport from source regions to Xi'an and Qingdao. This suggests that large amounts of dust transported from deserts to Xi'an and Qingdao do not undergo significant aging to increase aluminum solubility. Instead, local haze in Qingdao may enhance aluminum solubility, but this accounts for only a small fraction of annual dust, which is not the dominant component.

**Reply:** Indeed our work indicates that in spring when large amounts of mineral dust aerosols are transported to Qingdao, aerosol Al solubility does not increase a lot because the aging of mineral dust is very limited. However, since the focus of our manuscript is on the variation of aerosol Al solubility, we choose not to make this statement more explicitly. Modeling studies can give more quantitative results in this aspect.

20. Figure 5 should include the p-values of the fits. Has the fit passed significance testing? The correlation appears weak, making it difficult to support the authors' claim of an inverse relationship.

**Reply:** Compared to previous work which reported such inverse dependence, the inverse dependence we reported is quite obvious. We have further tried to fit this inverse dependence using the exponential function, because such quantitative relationship can be very useful for

modeling studies. Although the *r* values are not very large, the *p* values are all <0.01. In the revised manuscript (page 17), we have updated Figure 5 to include *p* values.

21. Line 344: On rapidly transported dust days, dust in Xi'an may exhibit external mixing with acidic components, whereas on haze days, over 95% of dust is internally mixed with acidic components. The authors could separately analyze aluminum solubility's relationship with acidic components on haze days and dust days to validate their explanation.

25. Line 410: I recommend categorizing samples into dust days and haze days to examine the impact of mixing state.

**Reply:** These two comments (No. 21 and 25) are closely related, and are therefore addressed together. We checked the dependence of aerosol Al solubility on acid species on haze and dust days separately, and there was still on correlation; one reason is that the numbers of haze and dust days were very small.

As suggested by the referee, we further examined the samples with low Al solubility, high RH and high acidity. These samples were mostly found in clear days, perhaps due to the influence of local resuspended dust for which chemical aging was very limited. In the revised manuscript (page 24), we have added one sentence to discuss these special samples: "We also note that samples with low Al solubility but high RH and high acidity were mostly found in clean days, perhaps due to the influence of local resuspended dust for which chemical aging was very limited."; moreover, I have added anotehr sentence to discuss future work which can provide further insights: "Single particle analysis which provides mixing state information can give further insights."

22. In Figure 6, "r" is used, while Figure 5 uses "r²," and Figure 6 includes p-values. It is recommended to standardize the plotting conventions.

**Reply:** As suggested, in the revised manuscript (page 17) we have updated Figure 5, and the new version uses "*r*" and includes *p* values.

23. Line 377: The analysis here focuses only on the relationship between local RH variations and aluminum solubility in Xi'an. If high RH promotes liquid-phase reactions that increase aluminum solubility, then aluminum solubility should also rise during high-RH conditions in Xi'an. The role of local chemistry should not depend on the distance from the desert.

24. Line 379: It is difficult to understand why high RH in Qingdao increases aluminum solubility but not in Xi'an. Atmospheric chemical processes should be similar—are there other controlling factors influencing this aluminum solubility mechanism?

**Reply:** These two comments (No. 23 and 24) are closely related, and are therefore addressed together. The different RH dependence of Al solubility at two two locations was due to the difference in aging extents. In the revised manuscript we have made the following two changes to make our explanation more explicit:

1) "This again may imply that chemical processing had very limited impact on aerosol Al solubility at Xi'an, as mineral dust particles mostly remained externally mixed with secondary species and their aging extent was very limited (Wang et al., 2014; Wu et al., 2017)." (page 21).

2) "In contrast, RH played an important role in regulating aerosol Al solubility at Qingdao, because mineral dust particles observed at Qingdao had been transported through the North China Plain and were substantially aged." (page 22)

26. Line 416: I disagree with labeling dust in Xi'an as "fresh dust," given that it originates over 3,000 km away.

**Reply:** In fact in our original manunscript (Line 414-416), we state that "aging extent of dust particles as rather limited at Xi'an", and "fresh" was used to describe dust samples used by Shi et al. (2011) who investigated topsoil samples collected over deserts. In order to avoid misundstanding, in the revised manuscript (page 24) we have changed "fresh" to "unaged".

---

## Author Comment (AC2)

Comments by referees are in blue.
Our replies are in black.
Changes to the manuscript are highlighted in red both here and in the revised manuscript.

**Reply to referee #1**

This manuscript provides a concise and informative comparison of aerosol aluminum (Al) solubility across seasons and sites (Xi'an and Qingdao), highlighting notable differences resulted from atmospheric aging. The use of < 1 μm and > 1 μm size fractions adds important size segregation to the analysis, and the interpretation grounded in source proximity and dust aging processes is scientifically sound and well-discussed.

**Reply:** We would like to thank ref #1 for reviewing our manuscript and recommending it for publication after minor revion. We have addressed these comments and updated the manuscript accordingly; when we do not quite agree with ref#1, we have provided proper explanation. Please find more details below.

Please see below for my detailed comments:

1. in the abstract, "Furthermore, seasonal variability of Al solubility, its correlation with relative abundance of sulfate and nitrate, and its dependence on relative humidity (RH), are all different at the two locations." it would be better if authors explain in detail how different these parameters are.

**Reply:** Indeed we would like to provide further details about aerosol Al slolubility in the abstract. However, the abstract cannot exceed 250 words, as required by the journal. As a result, we cannot provide these details in the abstract; instead, we present them in the conclusion.

2. authors used the phrase "significantly"/"significant" suggesting statistical analysis, but no mention is made of the statistical test used (e.g., t-test, ANOVA). Please clarify the method and significance level (e.g., $p < 0.05$) and indicate the results either in graphs in main text or in supplemental tables.

**Reply:** This is a good comment. Except correlation analysis, we did not use other statistical tests (e.g., t-tests); we use "significant/significantly" in our original manuscript mostly to express the extent, and do not indicate statistical significance. As a result, in the revised manuscript we have changed "significant" to "obvious", "great", or similar words, in order to avoid misunderstanding.

3. in line 468, authors said "Aerosol Al solubility at Xi'an showed no significant correlation with relative abundance of sulfate or nitrate", but in Table S8, the pearson r values were significant for > 1 μm particles at Xi'an in autumn and winter. please provide explanation.

**Reply:** The $r$ values for coarse particles at Xi'an in autumn and winter were mainly dictated by three outliers for which aerosol Al solubility was very high. If we excluded these three outliers, $r$ values decreased from 0.70 to 0.40 for autumn and from 0.93 to 0.44 for winter, and the correlations became insignificant ($p <0.05$). This further supports our original statement. In repsonce to this comment, we have modified Table S8 in the revised SI (page 9) to provide r values after excluding outliers, and please refer to our revised SI for further information.

4. in Figure 8, usually ascending order is used, such as "<2.5", "2.5-3.0"...

**Reply:** This is a good suggestion. In the revised manuscript (page 23) we have updated this figure, as suggested.

5. foggy weather might promote aluminum complexation reactions with organics. It's interesting that dissolved Al and Al solubility increased a lot in fog conditions in Table S7. Authors can consider adding a paragraph discussing this.

**Reply:** Indeed aerosol Al solubility was much higher during fog days, and in our original manuscript (page 15) we have used one paragrapgh to discuss this phenomenon. Acid and ligand processing can both enhance aerosol solubility, but at present it is difficult to disentangle their individual contributions. In the revised manuscript (page 16) we have added one sentence for further discussion: "Acid and ligand processing can both enhance aerosol Al solubility, although at present it is difficult to disentangle their individual contributions."

---

## Referee Report (RR1)

Thank you for the revisions and your response to my comments. However, I note that the authors have not made substantial modifications to the manuscript, nor have they adequately addressed my concerns regarding the scientific robustness of the paper's conclusions. Measuring the depth of analysis by page number is inappropriate; a paper should delve into scientific exploration deeply, even if it is concise. The core conclusion proposed by the authors—that atmospheric chemical processing alters the solubility of aluminum in aerosols—while not necessarily incorrect, lacks convincing support from the current explanations. The authors fail to rigorously demonstrate that this is a dominant factor influencing aerosol aluminum solubility.

Regarding the abstract, I maintain that the current version is overly general. I provided specific suggestions for improvement in my previous comments, yet the authors have made almost no changes to the abstract. I believe that carefully crafting the language to distill the core scientific information would not significantly increase the word count and could even make it more concise. The current abstract still lacks essential scientific evidence and in-depth quantitative analysis, making it unsuitable for a qualified research paper. Furthermore, the authors have not proofread this critical section carefully, as evident from the misaligned lines (between lines 36 and 37) and the presence of extra spaces or characters. I urge the authors to treat the revision process with greater seriousness.

I understand that the solubility of aluminum in dust deposited into the ocean can vary across different maritime regions and times, potentially significantly. However, this study only observes aluminum solubility at two terrestrial sites. The connection to the inference about oceanic dust deposition is not direct. Even if we clarify the spatiotemporal characteristics of aerosol aluminum solubility, how does that allow us to better constrain oceanic dust deposition? If the solubility of aluminum in dust varies greatly, how can we effectively use dissolved aluminum concentrations in seawater to constrain oceanic dust deposition? The logic behind this is unclear to me.

Concerning the issue of local resuspended dust, as the authors mentioned, its aluminum solubility is typically lower than that of desert dust, which is a consensus in many studies. However, the higher solubility observed in Qingdao compared to Xi'an does not automatically imply that local resuspended dust has a minimal influence in Qingdao. A more plausible explanation could be that emissions of local resuspended dust are much greater in Xi'an, thereby lowering the overall solubility there. In contrast,

Qingdao might have less local resuspended dust, resulting in a relatively higher observed solubility. This is not even the most critical point. The more crucial issue is that if the interference from local resuspended dust is substantial, the paper's conclusions regarding the properties and transport of desert dust cannot be explained clearly and rationally.

In my previous comment, I pointed out that desert dust rarely occurs in Xi'an during winter because the major dust sources in northern China are typically snow-covered, with frozen or moist soil that prevents dust emission even under strong winds. Therefore, the dust observed in Xi'an during winter is likely predominantly local resuspended dust. In their response, the authors shifted the focus by stating that many studies show dust is a significant component of aerosols in Xi'an. However, this refers to the conditions in spring, not winter.

If the authors hypothesize that the dust samples originate from the Loess Plateau, which is close to Xi'an, they must provide substantial evidence to support this claim. It is important to distinguish concepts clearly: the Loess Plateau is generally not considered a dust \*source\* region but rather a depositional area for aeolian dust. The primary dust sources affecting China are located in southern Mongolia and China's own deserts (e.g., Taklamakan, Badain Jaran, Tengger, and Kubuqi deserts). These source regions are almost all over a thousand kilometers away from Xi'an, not "quite close" as suggested.

Finally, regarding the authors' explanation for the smaller difference in aluminum solubility between the two cities in spring—attributing it to faster transport due to higher wind speeds, thus less aging—it is important to note that major dust events are typically associated with strong winds during transport from west to east. Does this imply that the solubility of aluminum is less affected during these significant dust events, which are precisely the events of greatest interest for transport and deposition into the oceans? This point requires further clarification.

---

## Author Response (AR2)

Dear Professor Guangjie Zheng,

Thank you very much for handling our manuscript submitted to ACP for the consideration of publication (manuscript number: egusphere-2025-2235; title: Atmospheric chemical processing dictates aerosol aluminum solubility: insights from field measurement at two locations in northern China).

The second version of our manuscript has been reviewed by two referees again. Ref #1 only has a few very minor comments, while ref #2 still has some major concerns. We have carefully addressed these comments and revised our manuscript accordingly. We believe that the revised manuscript can be accepted for publication, and highly appreciate these comments which have helped us further improve our work.

We would like to take this opportunity to thank you and the two referees for all the inputs. Please feel free to contact us if you need further information.

Dr. Mingjin Tang, Professor
Guangzhou Institute of Geochemistry
Chinese Academy of Sciences
Guangzhou 510640, China

Comments by referees are in blue.

Our replies are in black.

Changes to the manuscript are highlighted in red both here and in the revised manuscript.

**Reply to referee #1**

This manuscript presents a well-designed and thorough investigation of the spatial and seasonal variations in aerosol aluminum (Al) solubility, focusing on atmospheric aging by comparing solubilities at Xi'an and Qingdao. The authors integrated extensive field measurements, performed statistical analyses, and illustrated the effects of aging process in modulating Al solubility.

**Reply:** We would like to thank referee #1 for reviewing our manuscript again and recommending it for publication after minor revision. We have addressed his/her comments and revised the manuscript accordingly, as detailed below.

The manuscript would benefit from explaining the relatively large overlapping Al solubility data. The viewpoints are scientifically solid, however, the gap between the statistical analyses and conclusions requires more explanation. For example, in Figures 6 and S1 excluded outliers for Xi'an, but a very high solubility datapoint is kept for Qingdao instead and potentially makes the regression model significant.

**Reply:** Indeed Al solubility reported in our work show relatively large overlapping. This is unfortunately unavoidable for most (if not all) of field measurements, since environmental conditions are very complicated in the real atmosphere.

For Figures 6 and S1 in the previous versions of our manuscript, we excluded outliers at Xi'an but did not exclude outliers at Qingdao. We also carried out statistical analysis after excluding the outliers at Qingdao, and this almost led to no change. In the revised manuscript (page 21) we have made the following change to the caption of Figure 6: "...(d) supermicron particles at Qingdao (the *r* value changed from 0.81 to 0.74 if the data point with the highest Al solubility was excluded)." Similar change was also made to the caption of Figure S1 in the revised supplement (page 10): "...(d) supermicron particles at Qingdao (the *r* value changed from 0.81 to 0.77 if the data point with the highest Al solubility was excluded)." Below are technical corrections to be noticed:

1. Line 36, in the abstract, there is a new line that's not supposed to be there.

**Reply:** We would like to thank referee #1 for pointing out this error, which we should have avoided. In the second round of review, referee #2 insisted that the abstract was overly general. As a result, we have substantially modified the abstract in the revised manuscript (page 2); in addition, we have carefully checked the entire manuscript and supplement to avoid errors which we should avoid.

2. Legend in Figure 7 should have a box like other figures in the manuscript.

**Reply:** As suggested, we have updated Figures 7 (page 22) and 8 (page 23) in the revised manuscript.

**Reply to referee #2**

Thank you for the revisions and your response to my comments. However, I note that the authors have not made substantial modifications to the manuscript, nor have they adequately addressed my concerns regarding the scientific robustness of the paper's conclusions. Measuring the depth of analysis by page number is inappropriate; a paper should delve into scientific exploration deeply, even if it is concise. The core conclusion proposed by the authors—that atmospheric chemical processing alters the solubility of aluminum in aerosols—while not necessarily incorrect, lacks convincing support from the current explanations. The authors fail to rigorously demonstrate that this is a dominant factor influencing aerosol aluminum solubility.

**Reply:** We would like to thank referee #2 for reviewing our manuscript again. We tried to address the comments he/she raised in the first round of review. Since referee #2 still has some major concerns, we have carefully addressed these remaining concerns and revised the manuscript again, as detailed below. We highly appreciate these comments which have helped us significantly improve our work.

1. Regarding the abstract, I maintain that the current version is overly general. I provided specific suggestions for improvement in my previous comments, yet the authors have made almost no changes to the abstract. I believe that carefully crafting the language to distill the core scientific information would not significantly increase the word count and could even make it more concise. The current abstract still lacks essential scientific evidence and in-depth quantitative analysis, making it unsuitable for a qualified research paper. Furthermore, the authors have not proofread this critical section carefully, as evident from the misaligned lines (between lines 36 and 37) and the presence of extra spaces or characters. I urge the authors to treat the revision process with greater seriousness.

**Reply:** We would like to thank referee #2 for pointing out the error we made in the abstract, which we should have avoided. For the third version (the latest version) of our manuscript, we have carefully checked the entire manuscript and supplement to avoid errors which we should avoid

As referee #2 insisted that the abstract is overly general, we have decided to take his/her suggestion and revised the abstract: we have included core scientific information in the abstract, and deleted some non-critical words (in order not to exceed 250 words). Below is the updated abstract which can also be found in the revised manuscript (page 2): "Deposition of mineral dust aerosol into open oceans impacts marine biogeochemistry, and the deposition flux can be constrained using dissolved aluminum (Al) in surface seawater as a tracer. However, aerosol Al solubility, a critical parameter used in this method, remains highly uncertain. We investigated seasonal variations of aerosol Al solubility for supermicron and submicron particles at two locations (Xi'an and Qingdao) in northern China. Aerosol Al solubility was very low at Xi'an, showed no apparent variation with seasons or relative humidity, and was not correlated with sulfate or nitrate; in contrast, Al solubility was much higher at Qingdao, exhibited distinct seasonal variability, and increased with relative humidity and the abundance of sulfate and nitrate. All these features observed for Al solubility at the two locations can be explained by the effects of atmospheric chemical processing. Mineral dust transported to Xi'an (an inland city in Northwest China) was still not obviously aged and thus chemical processing had little effect on aerosol Al solubility; after arriving at Qingdao (a coastal city in the Northwest Pacific), mineral dust was substantially aged by chemical processing, leading to significant enhancement in aerosol Al solubility. Our work further reveals that aerosol liquid

water and acidity play vital roles in the dissolution of aerosol Al by atmospheric chemical processing. We suggest that chemical aging can lead to spatiotemporal variation of aerosol Al solubility, and this should be considered when using dissolved Al in surface seawater to constrain oceanic dust deposition."

2. I understand that the solubility of aluminum in dust deposited into the ocean can vary across different maritime regions and times, potentially significantly. However, this study only observes aluminum solubility at two terrestrial sites. The connection to the inference about oceanic dust deposition is not direct. Even if we clarify the spatiotemporal characteristics of aerosol aluminum solubility, how does that allow us to better constrain oceanic dust deposition? If the solubility of aluminum in dust varies greatly, how can we effectively use dissolved aluminum concentrations in seawater to constrain oceanic dust deposition? The logic behind this is unclear to me.

**Reply:** A good knowledge of spatiotemporal characteristics of aerosol Al solubility can inform us how to develop parameterizations of aerosol Al solubility, which can be used to better constrain dust deposition. To make it more explicit, we have made the following two changes in the revised manuscript.

- 1. Page 2: "We suggest that chemical aging can lead to spatiotemporal variation of aerosol Al solubility, and this should be considered when using dissolved Al in surface seawater to constrain oceanic dust deposition."
- 2. Page 4: "In order to better constrain the oceanic dust deposition using dissolved Al in seawater as a tracer, we need to develop parameterizations for aerosol Al solubility, and this requires spatiotemporal variability of aerosol Al solubility to be understood and processes and mechanisms which drive such variations to be elucidated."

Although our field observations were conducted only at two terrestrial sites, the results reveal that atmospheric chemical processes play an important role in controlling the variation of aerosol Al solubility. In the revised manuscript (page 27-28) we have added one sentence to discuss the implications and caveats of our work: "Although our measurements were only conducted at two sites, our work provides important insights into processes driving spatiotemporal variability of aerosol Al solubility, and such understanding can aid us to develop aerosol Al solubility parameterizations."

3. Concerning the issue of local resuspended dust, as the authors mentioned, its aluminum solubility is typically lower than that of desert dust, which is a consensus in many studies. However, the higher solubility observed in Qingdao compared to Xi'an does not automatically imply that local resuspended dust has a minimal influence in Qingdao. A more plausible explanation could be that emissions of local resuspended dust are much greater in Xi'an, thereby lowering the overall solubility there. In contrast, Qingdao might have less local resuspended dust, resulting in a relatively higher observed solubility. This is not even the most critical point. The more crucial issue is that if the interference from local resuspended dust is substantial, the paper's conclusions regarding the properties and transport of desert dust cannot be explained clearly and rationally.

**Reply:** We would like to point out that referee #2 may misunderstand what we stated. We did not state (previous studies did now show either) that Al solubility was lower for local resuspended dust than desert dust. In fact, Al solubility was always very low for soil and mineral dust samples examined in previous studies. In the revised manuscript (page 4) we have made the following change to make this clearer: "The initial Al solubility is generally low (typically <1.5%) for soil or mineral dust samples (Mulder et al., 1989; Duvall et al., 2008; Shi

et al., 2011; Aghnatios et al., 2014; Li et al., 2022)". Indeed we cannot exclude the contribution of local resuspended dust. This is why in the second version of our manuscript we changed "desert dust" to "mineral dust", in order not to exclude the contribution of local resuspended dust.

It is very likely that the contribution of local resuspended dust was lower in Qingdao than Xi'an. However, as Al solubility of local resuspended dust is not higher than desert dust, lower contribution of local resuspended dust can NOT explain either the much higher Al solubility (up to) observed at Qingdao or the dependence of Al solubility at Qingdao on RH and relative abundance of secondary species. In other words, we need to look for sources/processes which can enhance Al solubility. In Sections 3.2 and 4, we discussed several possibilities and came to the conclusion that atmospheric chemical processing dictates aerosol aluminum solubility.

4. In my previous comment, I pointed out that desert dust rarely occurs in Xi'an during winter because the major dust sources in northern China are typically snow-covered, with frozen or moist soil that prevents dust emission even under strong winds. Therefore, the dust observed in Xi'an during winter is likely predominantly local resuspended dust. In their response, the authors shifted the focus by stating that many studies show dust is a significant component of aerosols in Xi'an. However, this refers to the conditions in spring, not winter.

**Reply:** Besides spring, Asian dust events also occur in winter. On the other hand, we agree with referee #2 that local resuspended dust can also play a significant role; this is why in the second version we used "mineral dust" instead of "desert dust", in order not to exclude local resuspended dust. In the revised manuscript (Page 10) we have made the following change to provide further clarification: "Furthermore, besides spring, Asian dust also occurs in winter (Cai et al., 2020; Wang et al., 2020), and a previous study (Huang et al., 2014) suggested that the dust-related source, including local resuspended dust, contributed 56% to PM2.5 during a severe haze event at Xi'an."

5. If the authors hypothesize that the dust samples originate from the Loess Plateau, which is close to Xi'an, they must provide substantial evidence to support this claim. It is important to distinguish concepts clearly: the Loess Plateau is generally not considered a dust \*source\* region but rather a depositional area for aeolian dust. The primary dust sources affecting China are located in southern Mongolia and China's own deserts (e.g., Taklamakan, Badain Jaran, Tengger, and Kubuqi deserts). These source regions are almost all over a thousand kilometers away from Xi'an, not "quite close" as suggested.

**Reply:** The Loess Plateau is a depositional region for Asian dust, but it is also an active source of Asian dust. In the revised manuscript (page 6) we have made the following change to clarify this: "Xi'an is an inland city in northwestern China, located at the southern edge of the Loess Plateau which is also an active source of mineral dust (Cao et al., 2008; Jeong, 2020; Haugvaldstad et al., 2024), and the aging extent of mineral dust at Xi'an was found to be quite limited (Wang et al., 2014; Wu et al., 2017)."

We fully agree that some dust sources are quite far from Xi'an, and the aging of dust particles transported to Xi'an is rather limited mainly because anthropogenic emission in Northwest China is much smaller. As a result, compared to the first version, we have made the following change in the second version (line 251-255, page 13, the third/current version): "There are several important dust sources in Northwest China, being far from (up to a few thousand km) or close to Xi'an. More importantly, anthropogenic emission in Northwest China is much smaller than the North China Plain, and thus the aging extent of mineral dust transported to Xi'an was rather limited (Wang et al., 2014; Wu et al., 2017)." Moreover, the original sentence in the second version "Xi'an is an inland city in northwestern China, and the

aging extent of dust was found to be quite limited at Xi'an due to its proximity of desert regions (Wang et al., 2014; Wu et al., 2017)" has been changed in the revised manuscript (page 6) to "Xi'an is an inland city in northwestern China, located at the southern edge of the Loess Plateau which is also an active source of mineral dust (Cao et al., 2008; Jeong, 2020; Haugvaldstad et al., 2024), and the aging extent of mineral dust at Xi'an was found to be quite limited (Wang et al., 2014; Wu et al., 2017)."

6. Finally, regarding the authors' explanation for the smaller difference in aluminum solubility between the two cities in spring—attributing it to faster transport due to higher wind speeds, thus less aging—it is important to note that major dust events are typically associated with strong winds during transport from west to east. Does this imply that the solubility of aluminum is less affected during these significant dust events, which are precisely the events of greatest interest for transport and deposition into the oceans? This point requires further clarification.

**Reply:** Indeed our work implies that the enhancement of aerosol Al solubility at Qingdao is limited during large dust events when large amounts of dust was emitted and despoited into the ocean. However, this does not necessarily imply that aerosol Al solubility remains low when dust particles are further transported to the open oceans, as Qingdao is a coastal site. To discuss this issue, we have added the following sentence in the revised manuscript (page 27-28): "Our work implies that during large dust events increase in aerosol Al solubility may be rather limited when dust is transported to Qingdao; nevertheless, when dust is transported further eastward to the open ocean, atmospheric chemical processing may substantially increase aerosol Al solubility."

---

## Author Response (AR3)

Dear Professor Guangjie Zheng,

Thank you very much for handling our manuscript submitted to ACP for the consideration of publication (manuscript number: egusphere-2025-2235; title: Atmospheric chemical processing dictates aerosol aluminum solubility: insights from field measurement at two locations in northern China).

The third version of our manuscript has been reviewed by two referees again. One referee has no further comments, and the other referee only has a minor comment on our abstract. We have addressed this comment and revised our abstract accordingly, and believe that the revised manuscript can be accepted for publication

We would like to take this opportunity to thank you and the two referees for all your very valuable inputs. Please feel free to contact us if you need further information.

Dr. Mingjin Tang, Professor

Guangzhou Institute of Geochemistry

Chinese Academy of Sciences

Guangzhou 510640, China

Comments by referees are in blue.

Our replies are in black.

Changes to the manuscript are highlighted in red both here and in the revised manuscript.

**Reply to referee #2**

This manuscript investigates seasonal and size-resolved fluctuations in aerosol aluminum (Al) solubility at two sites in northern China, inland Xi'an and coastal Qingdao. I would suggest adding numerical statements in the abstract. The current version lacks statistical comparison and provided only text statements. Specifically, lines 36-37: "substantially aged" and "leading to significant enhancement in aerosol Al solubility". The authors should add numbers to support their conclusions.

**Reply:** We would like to thank referee #1 for reviewing our manuscript and recommending it final publication after minor revision to the abstract.

As suggested, in the revised manuscript (page 2) we have made the following changes to provide numerical information for aerosol Al solubility at the two locations: "Aerosol Al solubility was very low at Xi'an (0.11-9.1%), showed no apparent variation with seasons or relative humidity, and was not correlated with sulfate or nitrate; in contrast, it was much higher at Qingdao (0.06-23.4%), exhibited distinct seasonal variability, and increased with relative humidity and the abundance of sulfate and nitrate."

In addition, we have deleted some words in the abstract in order to keep the number of words below 250. Please kindly refer to the revised manuscript for more information.